# A Pilot Study of Wet Lung Using Lung Ultrasound Surface Wave Elastography in an Ex Vivo Swine Lung Model

**Xiaoming Zhang** [1,*], **Boran Zhou** [1] and **Alex X. Zhang** [2]

1    Department of Radiology, Mayo Clinic, Rochester, MN 55905, USA
2    Department of Biochemistry and Molecular Biology, Mayo Clinic, Rochester, MN 55905, USA
*    Correspondence: Zhang.xiaoming@mayo.edu; Tel.: +507-538-1951; Fax: +507-266-0361

**Abstract:** Extravascular lung water (EVLW) is a basic symptom of congestive heart failure and other conditions. Computed tomography (CT) is standard method used to assess EVLW, but it requires ionizing radiation and radiology facilities. Lung ultrasound reverberation artifacts called B-lines have been used to assess EVLW. However, analysis of B-line artifacts depends on expert interpretation and is subjective. Lung ultrasound surface wave elastography (LUSWE) was developed to measure lung surface wave speed. This pilot study aimed at measureing lung surface wave speed due to lung water in an ex vivo swine lung model. The surface wave speeds of a fresh ex vivo swine lung were measured at 100 Hz, 200 Hz, 300 Hz, and 400 Hz. An amount of water was then filled into the lung through its trachea. Ultrasound imaging was used to guide the water filling until significant changes were visible on the imaging. The lung surface wave speeds were measured again. It was found that the lung surface wave speed increases with frequency and decreases with water volume. These findings are confirmed by experimental results on an additional ex vivo swine lung sample.

**Keywords:** lung ultrasound surface wave elastography (LUSWE); lung water; surface wave speed; ex vivo swine lung

## 1. Introduction

Extravascular lung water (EVLW) is common for patients with congestive heart failure and other inflammatory conditions, such as acute respiratory distress syndrome [1]. Chest X-ray or chest computed tomography are standard methods used to assess EVLW in a clinic, but they require ionizing radiation and radiology facilities. Lung ultrasound reverberation artifacts called B-lines have been used to assess EVLW. However, B-line artifact analysis depends on expert interpretation and is subjective [2]. Lung ultrasound surface wave elastography (LUSWE) was developed [3–6] to measure the surface wave speed of a lung and was applied for assessing patients with interstitial lung disease (ILD) [7,8]. We studied a lung phantom sponge model using LUSWE to measure the water effects on the surface wave speed of sponges [9–11]. However, we could not identify the trend of surface wave speed with water in the sponge phantom model. This pilot study aimed at measuring the effect of lung water on the surface wave speed in an ex vivo swine lung model.

## 2. Methods

In the ex vivo lung model, a 0.1 s harmonic vibration was generated on the surface of the lung using a small vibrator. The surface wave propagation on the lung was measured using an ultrasound probe. The vibration excitation signal was generated using a function generator (Model 33120A, Agilent Inc., Santa Clara, CA, USA). The signal was then amplified using a power amplifier (Model PYLE PRO

PCA4, PYLE PRO Service Center, 1600 63rd Street, Brooklyn, NY 11204, USA). The excitation signal was used to excite a small vibrator shaker (Model: FG-142, Labworks Inc., Costa Mesa, CA 92626, USA). The shaker generated a local small vibration on the surface of lung using an indenter with a plastic ball with a diameter of 5 mm.

A Verasonics ultrasound system (Verasonics, Inc., Kirkland, WA 98034, USA) was used to acquire the ultrasound data. An ultrasound probe with a central frequency of 6.4 MHz L11-5 was used in the experiments. The in-phase/quadrature (IQ) data of ultrasound signals were analyzed by demodulation of the radio-frequency (RF) data of the ultrasound echo. The particle velocity $v$ due to the external vibration generation at a pixel in the axial direction of ultrasound beam was analyzed. $v$ was calculated using an autocorrelation method [12,13]. High frame imaging was obtained using the plane-wave transmission technique. Pulse repetition frequency (PRF) was 2000 pulses per second. Central frequency ($f$) was 6.4 MHz. Three pixels in the axial direction and two sampling points in the slow time direction were averaged for data processing. Prior to window selection, the whole autocorrelation matrix for each lateral direction was calculated as follows:

$$v = \frac{c}{4\pi f T_s} \left\{ \frac{\sum_{m=0}^{M-1} \sum_{n=0}^{N-2} [Q(m,n)I(m,n+1) - I(m,n)Q(m,n+1)]}{\sum_{m=0}^{M-1} \sum_{n=0}^{N-2} [I(m,n)I(m,n+1) + Q(m,n)Q(m,n+1)]} \right\} \tag{1}$$

where the pulse repetition period is $T_s = \frac{1}{PRF}$, c is the speed of sound of tissue and is assumed to be 1540 m/s, $I(m,n)$ and $Q(m,n)$ are the real and imaginary *IQ* date at a pixel ($m$, $n$). A $3 \times 3$ pixel spatial median-filter was used to improve the imaging.

In this ex vivo lung model, the surface wave speed was measured by analyzing the surface lung motion among several locations. $v_1(t)$ and $v_2(t)$ represent the tissue motion at two locations on the lung surface. The cross-spectrum $S(f)$ of two signals $v_1(t)$ and $v_2(t)$ at the excitation frequency f was analyzed by [14]:

$$S(f) = S_1^*(f) \cdot S_2(f) = \left| S_1(f) \cdot S_2(f) \right| \cdot e^{-j\Delta\varphi(f)} \tag{2}$$

where $S_1(f)$ and $S_2(f)$ are the Fourier transforms of $v_1(t)$ and $v_2(t)$, respectively; * denotes the complex conjugate; and $\Delta\varphi(f)$ is the phase change between $v_1(t)$ and $v_2(t)$ over distance between the two locations.

The change of surface wave phase with distance was analyzed to calculate the surface wave speed:

$$c_s(f) = 2\pi f |\Delta r / \Delta\varphi| \tag{3}$$

where $\Delta r$ is the distance between the two locations. The estimation of surface wave speed can be improved by measuring the phase change over multiple locations [15].

## 3. Experimental Design

Figure 1 shows the experimental design for testing the surface wave speed of a fresh ex vivo swine lung. The fresh lung was obtained from an approved Institutional Animal Care and Use Committee (IACUC) protocol after the swine was sacrificed. The swine lung was positioned on a testing table. A thick rubber material plate was between the lung tissue and the table to reduce wave reflection. The water was injected to the lung using a plastic tube connected to the trachea of lung. Figure 2a shows an example of a B-mode image of the lung. In this study, the surface lung motion was measured at eight locations. The surface wave speed was analyzed by the wave phase delay, relative to the first location. Figure 2b shows an example of wave speed at 100 Hz for the lung. The surface wave speed was 1.41 ± 0.06 m/s for the lung. Five measurements were performed at each frequency. The surface wave speed was analyzed in the format of mean ± SD for the five measurements. The surface wave speed was measured at 100 Hz, 200 Hz, 300 Hz and 400 Hz. The 100 Hz excitation signal was stronger than higher frequencies. The higher frequency waves have a smaller wave length but decay more rapidly [16]. The lung was tested at both the right and left lobes.

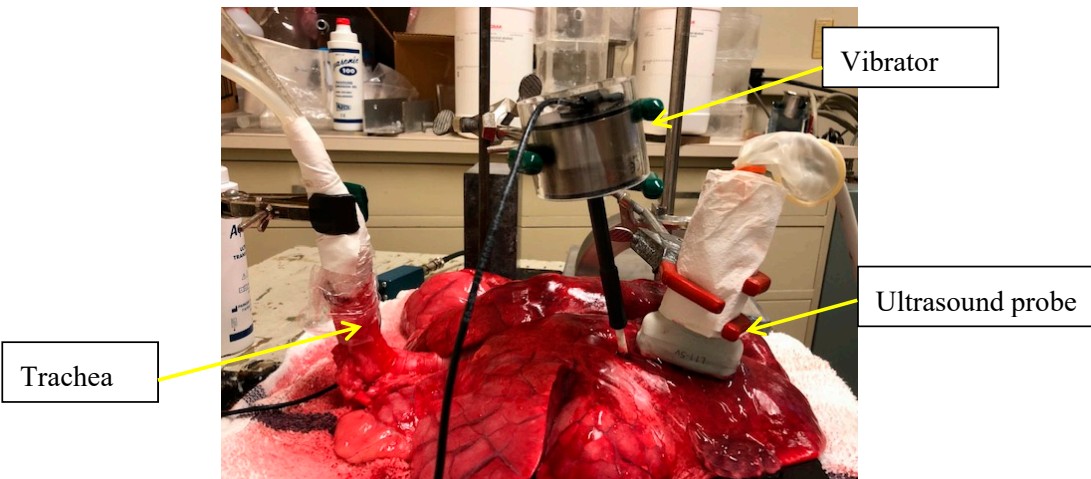

**Figure 1.** Experimental design for testing the surface wave speed on an ex vivo fresh swine lung. The swine lung was tested on a table. A thick rubber pad was placed between the lung and the table to reduce wave reflection. The water was injected into the lung through a plastic tube connected to the trachea of lung. In this ex vivo swine lung study, the surface wave propagation was generated using a small vibrator and measured using an ultrasound probe.

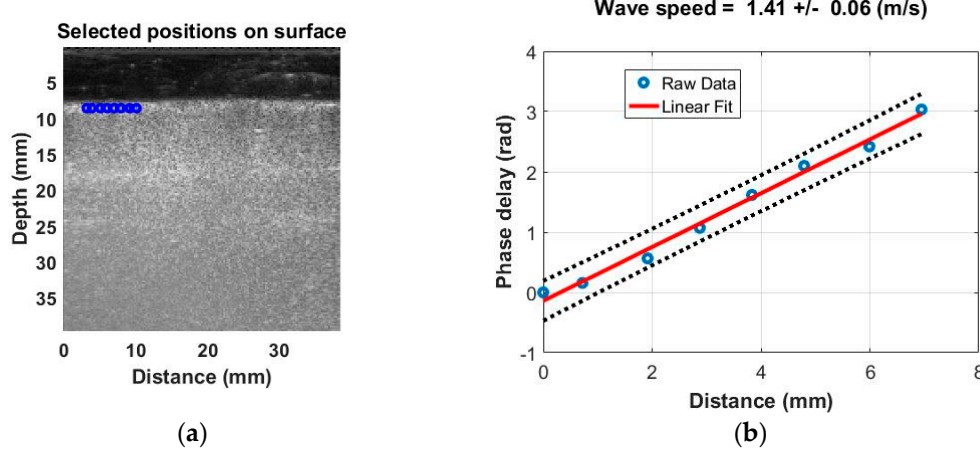

**Figure 2.** (**a**) An example of a B-mode image of the right lung of the swine. The lung motion was measured at eight locations on the surface of the lung. (**b**) An example analysis of the surface wave speed at 100 Hz. The lung surface wave speed was analyzed using the wave phase delay relative to the first location.

The lung was first tested at the baseline (no water injection) and at one lobe of the lung. Five measurements were performed at 100, 200, 300, and 400 Hz. The other lobe of the lung was then measured. After both lobes of the lung were tested, 120 mL of tap water was injected into the lung through its trachea. Ultrasound imaging was used to guide the water filling until significant changes were visible on the imaging. The lung surface wave speeds were measured at both lobes of the lung. Then, an additional 120 mL of water was filled into the lung. Both lobes of lung were measured again.

## 4. Results

Figure 3a–c show the surface wave speed of the right lobe of lung versus frequency for the water volumes 0 mL, 120 mL, and 240 mL, respectively. The surface wave speed increased with frequency, although there was a little drop from 200 Hz to 300 Hz for the water volume of 120 mL. In general, this is consistent with our experimental results on various tissues. Figure 4a–d show the surface wave speed of the right lobe of lung versus water volume at 100 Hz, 200 Hz, 300 Hz, and 400 Hz, respectively.

For the same lung, the surface wave speed of the left lobe of lung decreased with the water volume at 100 Hz, 200 Hz, 300 Hz, and 400 Hz. New experiments were performed on another ex vivo swine lung. The lung surface wave speeds were first measured at 100 Hz, 200 Hz, 300 Hz, and 400 Hz. The lung surface wave speeds of the lung were then measured again with 180 mL and 360 mL injected tap water. Figure 5a–d show the surface wave speed of the lung versus water volume at 100 Hz, 200 Hz, 300 Hz, and 400 Hz, respectively. Figure 5 shows that the lung surface wave speed decreases with the water volume.

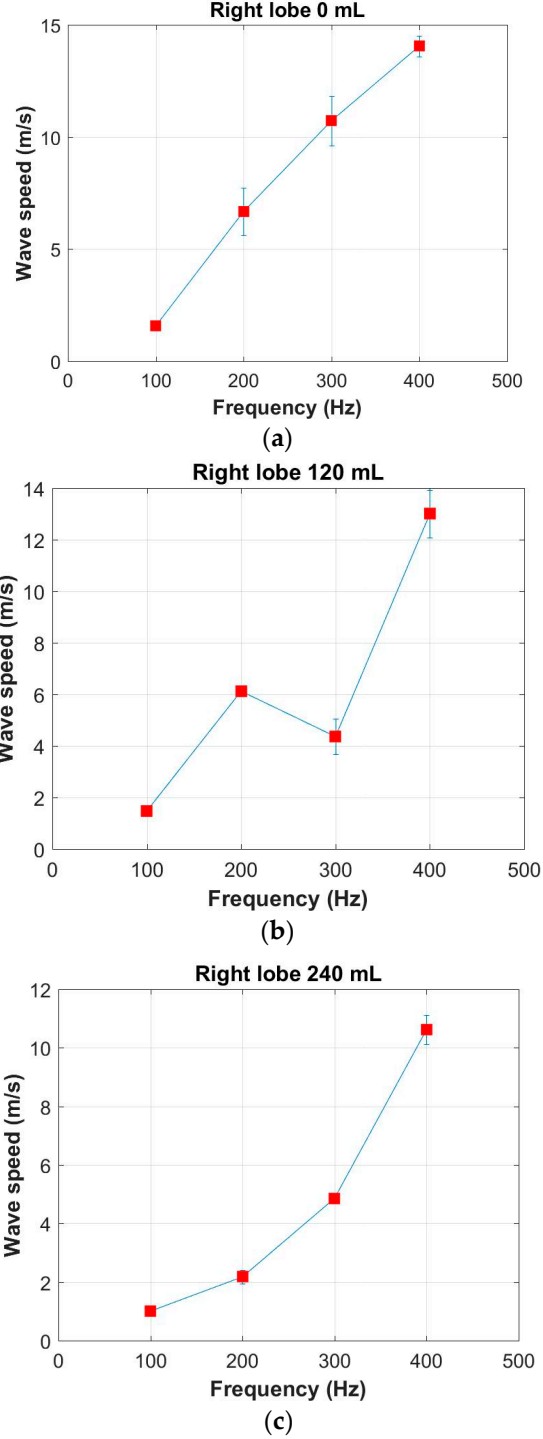

**Figure 3.** Measurements of surface wave speeds with frequency on the right lobe of the lung. (**a**) 0 mL, (**b**) 120 mL, and (**c**) 240 mL.

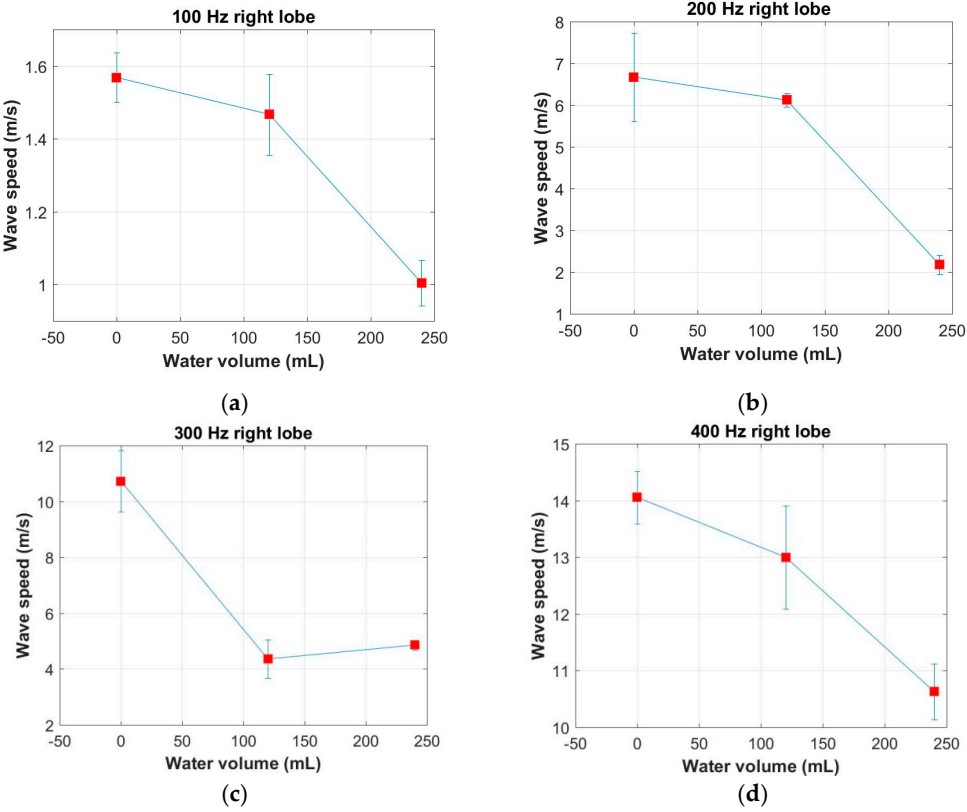

**Figure 4.** Measurements of surface wave speeds with water volume on the right lobe of the lung. (**a**) 100 Hz, (**b**) 200 Hz, (**c**) 300 Hz, and (**d**) 400 Hz.

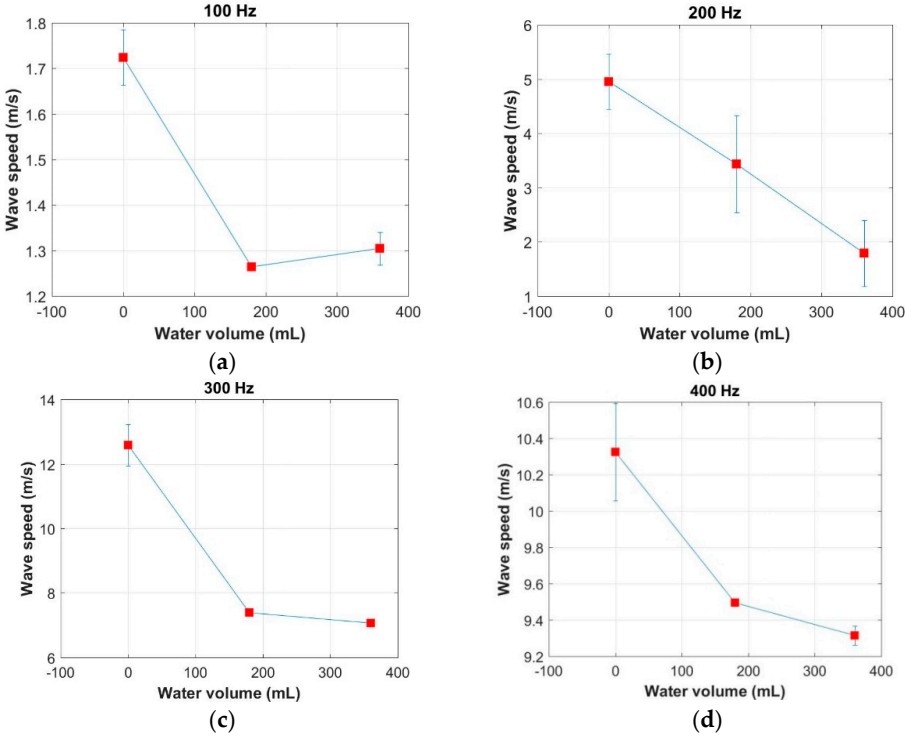

**Figure 5.** Measurements of surface wave speeds with water volume on another sample of an ex vivo swine lung. (**a**) 100 Hz, (**b**) 200 Hz, (**c**) 300 Hz, and (**d**) 400 Hz.

## 5. Discussion

The purpose of this study was to evaluate if water in the lung would affect the surface lung stiffness in an ex vivo swine lung model. In this pilot study, the swine lung is chosen because the swine lung is relatively large and easy to use for technique development. Mice lungs are very small and more difficult to use for technique developments. However, we will be cautious to use the swine lung data for humans because their material properties may be different. The tap water was injected to the lung through the trachea of the lung. Previously, we studied the change of lung surface wave speed with pulmonary pressure [17]. The pulmonary pressure was controlled by pumping the air through the trachea of lung. The pulmonary pressure was monitored using a pressure sensor. In this pilot study on lung water, the lung was tested at the baseline. Then some water was injected to the lung through its trachea. Ultrasound imaging was used to guide the water filling until significant changes were visible in the imaging. For future research, the quantification of water volume needs to be improved. In this study we quantified the water volume based on injection volume. We may also quantify lung water by measuring the water pressure of the lung. Injection of water through trachea is easy and feasible. However, the extravascular lung water for patients with lung edema is mainly due to fluid accumulation in the lung interstitium [18]. We will study the effects of induced lung water by injecting water through the pulmonary artery in future studies.

The lung surface wave speeds were measured using the wave phase delay technique. Therefore, the wave speed measurement was local and independent of the location of excitation. The lung surface wave speeds were tested at 100 Hz, 200 Hz, 300 Hz and 400 Hz. Viscoelasticity of the lung can be analyzed using the wave speed dispersion curve if the lung mass density is known. We will continue to develop a technique to estimate lung mass density [19].

It is found that lung surface wave speed decreases with water volume for both lobes of the lung, and at the four frequencies for this ex vivo swine lung model. These findings were confirmed by experimental results on another ex vivo swine lung sample. In a previous sponge phantom model, we could not identify the trend of wave speed with water in the sponge phantom model [10]. In the sponge phantom model, the change of mass density of phantom could be measured. Using the viscoelastic model, the shear viscosity of the sponge increased with water content and shear elasticity exhibited a subtle increase. We will try to quantify the lung mass density in ex vivo swine lungs so that we can estimate viscoelasticity of the lung, based on the wave speed measurements at the four frequencies.

## 6. Conclusions

In this pilot study, an ex vivo swine lung model was developed to measure the change of lung surface wave speed due to lung water. An ex vivo fresh swine lung was tested at both lobes of the lung. The lung surface wave speeds were measured at 100 Hz, 200 Hz, 300 Hz, and 400 Hz. Then, an amount of water was filled into the lung through its trachea. Ultrasound imaging was used to guide the water filling until significant changes were visible in the imaging. The lung surface wave speeds were measured again. The lung surface wave speed increased with frequency but decreased with water volume. These findings are confirmed by experimental results on another ex vivo swine lung sample.

**Author Contributions:** X.Z. conceptualized the research; X.Z., B.Z. and A.X.Z. performed the experiments. X.Z. analyzed the experimental data and wrote the manuscript. B.Z. and A.X.Z. helped with data analyses and revised the manuscript.

**Funding:** This study is supported by NIH R01HL125234 from the National Heart, Lung, and Blood Institute.

**Conflicts of Interest:** There are no conflicts of interest.

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
