# Peer review of "A Pilot Study of Wet Lung Using Lung Ultrasound Surface Wave Elastography in an Ex Vivo Swine Lung Model"

_applsci, doi:10.3390/app9183923_

Round 1
Reviewer 1 Report
In their paper "A pilot study of wet lung using lung ultrasound surface wave elastography in an ex vivo swine lung modes" the authors tested whether surface wave elastographic methods would allow to reliably identify extra vascular water inside the lungs. In the presented ex vivo pilot study using a swine lung the authors could show that mechanical waves directly applied to the outer surface of the lung change their travel speed dependent upon the water contained in the extra vascular alveolar space. Hammering with the fingers on the patients chest and back and listening to the sounding of the chest and the lungs especially using a stethoscope is a very simple and well established first clinical and medical test for identifying whether and how much the lung is blocked by water or other secrets. Given this fact the method presented by the authors seems to be quite promising to become the next generation of this quick and easy test allowing to identify the filling of the lung with water at early stages where stethoscope and human ear tests might still be inconclusive.
In general as far as i can judge from my background the presented research is conclusive and given the short space well described. I still would suggest the following improvements to the manuscript.
A) Restructure as follows
Introduction, Methods, Experimental design, Results, Discussion and conclusion.
and merge all descriptions which concern experimental design and setup split between the last paragraph of Methods section and the first paragraph of the Results section as well as the first paragraph of the Discussion within the Experimental design section.
B) Describe your results more elaborate. Given the short space of about eight pages I'd suggest to present plots for the left lobe only if they greatly differ from the ones of the right and thereby provide additional insight and information not conclusively deduceable form the plots of the right lobe or the left lobe shows different behavior which is not some not yet fully explainable experimental effect (right at 300Hz left at 400Hz?). For showing that the method works on both lobes and thus is independent of the size of the lung lobe i consider a single plot at max two plots from tow is sufficient.
Take care not mixing your findings with description of experiment as is now. Try to avoid duplication of experimental design description. In case needed refer to the description to present the identified changes causally related to the increased filling of the extra atrial alveolar space. etc.
C) Put discussion related to your findings like speculating about why the right lobe behaves differently at 300 Hz and the left at 400Hz. In the discussion section. Discuss other rather unexpected behavior of the data, is it caus of the relative difficulty to simulate a constant and steady filling, the not so optimal setup, actuator, sensor placements etc what ever most likely could an explanation to some unexpected findings you want to present in this publication.
D) Present the findings which directly influence the direction of your ongoing and near future research work in this direction in the outlook section and again avoid simply repeating the experimental design.
Author Response
Special thanks to you for your professional and valuable comments. Please see the attached file.

Reviewer 2 Report
Dear Authors,
I reviewed your manuscript, "A pilot study of wet lung using lung ultrasound surface wave elastography in an ex vivo swine lung model." The paper is very concisely written, with a clear explanation of the experimental methods and results. The results are interesting and I suggest acceptance; however, I do have a few minor comments.
1. I believe the manuscript could be strengthened with a bit more detail in the introduction. Please consider addressing the following:
a. What is known previously; what studies have been performed
b. What gap in knowledge this study specifically aims to fill or inspire additional attention
c. Why the choice of swine lung is appropriate, e.g., any limitations or reasons to think their materials properties may be different than humans?
2. It appears your study used one fresh swine lung for the entire study. Is this correct? If not, please state any differences in morphology between different cases. If only one lung was used, could you comment on whether repeated experiments could potentially give rise to false artifacts in data due to repeated use?
3. Since Figure 1 illustrates the experimental setup, could you please label the different aspects for visual clarity.
4. If possible, please move the legend in Figure 2(b) so that it does not cover some of the data.
5. It may be worthwhile to explicitly state what type of water you used in the experiments (tap, distilled, etc.)
Author Response

(The authors gave the same response as above.)

Round 2
Reviewer 1 Report
In their paper "A pilot study of wet lung using lung ultrasound surface wave
elastography in an ex vivo swine lung modes" the authors tested whether surface wave
elastographic methods would allow to reliably identify extra vascular water inside the
lungs. In the presented ex vivo pilot study using a swine lung the authors could show
that mechanical waves directly applied to the outer surface of the lung change their
travel speed dependent upon the water contained in the extra vascular alveolar space.
Hammering with the fingers on the patients chest and back and listening to the
sounding of the chest and the lungs especially using a stethoscope is a very simple and
well established first clinical and medical test for identifying whether and how much the
lung is blocked by water or other secrets. Given this fact the method presented by the
authors seems to be quite promising to become the next generation of this quick and
easy test allowing to identify the filling of the lung with water at early stages where
stethoscope and human ear tests might still be inconclusive.
In general as far as i can judge from my background the presented research is
conclusive and given the short space well described.
Given the fact that this publication describes first results obtained from a small feasibility and pilot study, they managed by addressing all comments from the first review to improve the presentation of their work done and achieved so far greatly.
This manuscript is a resubmission of an earlier submission. The following is a list of the peer review reports and author responses from that submission.
Round 1
Reviewer 1 Report
On pathophysiological basis the concept of « lung water» requires some specification. The amount of water in the lung is strictly controlled by restricting its filtration from the blood capillaries and by having an efficient draining system operated by lymphatics. So, extravascular lung water is interpreted as its accumulation in the lung interstitium (the space permeable to water between blood capillary walls and the alveolar barrier) or in the alveoli due to damage of the alveolar wall. The degree of a perturbation in lung fluid balance is key factor of importance at diagnositc level. In fact, the assessment of a minor increase in extravascular water (less than 10%) would be very helpful to prevent development towards a severe degree of lung edema.
The introduction, the discussion and the reference list suggest a considerable lack of knowledge on pathophysiology of lung water.
May I suggest a good rewiev on the control of lung fluid balance ; « Miserocchi G. Mechanisms controlling the volume of pleural fluid and extravascular lung water Eur Respir Rev 2009; 18: 114, 244–252. «
My main comment to this paper is that intratracheal injection of water is a model totally inappropriate to mimic any phase of pathophysiology of lung edema. It only mimics the case of drowning. Given this, I believe it might represent a pilot study to validate lung ultrasound surface wave elastography (LUSWE) as a tool of potential clinical use.
Reviewer 2 Report
The authors apply the technique developed earlier in their works [3-6] to a non-invasive detection of extravascular lung water. The technique is based on measuring the speed of low frequency (100-400Hz) surface wave in lungs. The work is pure experimental. The authors studied dependence of the surface wave speed on the amount of water in ex vivo swine lung. They found that the speed decreases noticeably with the increase of water in the lung.
The work could be useful for medical applications but I do not see any its essential contribution into physical or engineering sciences compared to [3-6]. Therefore some additional analysis should be included in order to be publishable in the Journal.
Minor remark:
Figure 3. The x-coordinate is not indicated in the plots. It is described in the text: “x coordinate represents the baseline, water injection, and additional 120 ml water injection with values of 1, 2, and 3”. Then numbers 1,2,3 could be associated with those cases, but then values 0.5, 1.5,2.5,3.5 can confuse the readers and should be removed. The better way might be to indicate the amount of water added. The best way is to relate the amount of water with the EVLW severity then the result could be more valuable for the medical community.